# Invasive Fungal Disease After Chimeric Antigen Receptor-T Immunotherapy in Adult and Pediatric Patients

**DOI:** 10.3390/pathogens14020170

**Published:** 2025-02-08

**Authors:** Paschalis Evangelidis, Konstantinos Tragiannidis, Athanasios Vyzantiadis, Nikolaos Evangelidis, Panagiotis Kalmoukos, Timoleon-Achilleas Vyzantiadis, Athanasios Tragiannidis, Maria Kourti, Eleni Gavriilaki

**Affiliations:** 12nd Propedeutic Department of Internal Medicine, Hippocration Hospital, Aristotle University of Thessaloniki, 54642 Thessaloniki, Greece; pascevan@auth.gr (P.E.); evangeln@auth.gr (N.E.); kalmoukosp@yahoo.gr (P.K.); 2Children & Adolescent Hematology-Oncology Unit, Second Department of Pediatrics, School of Medicine, Aristotle University of Thessaloniki, 54124 Thessaloniki, Greece; konstantinos.tragiannidis@gmail.com (K.T.); atragian@hotmail.com (A.T.); makourti@icloud.com (M.K.); 3Department of Microbiology, Medical School, Aristotle University of Thessaloniki, 54124 Thessaloniki, Greece; athanvyzanti@gmail.com (A.V.); tavyz@auth.gr (T.-A.V.); 4Hematology Department and Bone Marrow Transplant (BMT) Unit, G. Papanicolaou Hospital, 57010 Thessaloniki, Greece

**Keywords:** antifungal prophylaxis, chimeric antigen receptor-T, cytokine release syndrome, fungal, immune effector cell-associated neurotoxicity syndrome, invasive fungal disease

## Abstract

Invasive fungal diseases (IFDs) have been documented among the causes of post-chimeric antigen receptor-T (CAR-T) cell immunotherapy complications, with the incidence of IFDs in CAR-T cell therapy recipients being measured between 0% and 10%, globally. IFDs are notorious for their potentially life-threatening nature and challenging diagnosis and treatment. In this review, we searched the recent literature aiming to examine the risk factors and epidemiology of IFDs post-CAR-T infusion. Moreover, the role of antifungal prophylaxis is investigated. CAR-T cell therapy recipients are especially vulnerable to IFDs due to several risk factors that contribute to the patient’s immunosuppression. Those include the underlying hematological malignancies, the lymphodepleting chemotherapy administered before the treatment, existing leukopenia and hypogammaglobinemia, and the use of high-dose corticosteroids and interleukin-6 blockers as countermeasures for immune effector cell-associated neurotoxicity syndrome and cytokine release syndrome, respectively. IFDs mostly occur within the first 60 days following the infusion of the T cells, but cases even a year after the infusion have been described. *Aspergillus* spp., *Candida* spp., and *Pneumocystis jirovecii* are the main cause of these infections following CAR-T cell therapy. More real-world data regarding the epidemiology of IFDs and the role of antifungal prophylaxis in this population are essential.

## 1. Introduction

Chimeric antigen receptor-T (CAR-T) immunotherapy has revolutionized the management of patients with relapsed/refractory (R/R) B cell malignancies and has become the standard of care [1]. Up to date, four CAR-T cell products have been approved for the treatment of B cell lymphomas and B-acute lymphoblastic leukemia (B-ALL): axicabtagene ciloleucel for diffuse large B cell lymphoma (DLBCL), primary mediastinal large B cell lymphoma (PMBCL), and follicular lymphoma (FL); brexucabtagene autoleucel for R/R mantle cell lymphoma (MCL) and B-ALL; lisocabtagene maraleucel for DLBCL, PMBCL, and FL; and tisagenlecleucel for DLBCL and B-ALL [2]. Moreover, two CAR-T cell products are available for patients with R/R multiple myeloma (M/M): idecabtagene vicleucel and ciltacabtegene autoleucel [3]. Despite the therapeutic efficacy of this treatment approach, various complications are observed in the post-infusion period [4].

Cytokine release syndrome (CRS) and immune effector cell-associated neurotoxicity syndrome (ICANS) are considered among the major toxicities developed in CAR-T cell recipients early (within 10 days) after the CAR-T cell immunotherapy administration [5,6]. Endothelial injury and immune dysregulation are implicated in the pathogenesis of these syndromes [7,8]. Tocilizumab, a monoclonal antibody targeting interleukin 6 (IL-6), and corticosteroids are used for the management of patients who develop CRS [9,10]. Prolonged hematological toxicity (anemia, thrombocytopenia, and neutropenia) can also be observed in the post-infusion period [11]. Various scores have been developed and validated for the prediction of CAR-T cell-related toxicity onset and outcomes in these patients [12].

Prolonged neutropenia and lymphopenia, prior chemotherapy, history of hematopoietic cell transplantation (HCT), coexisting comorbidities, and administration of corticosteroids and tocilizumab result in a net state of immunosuppression [13]. Infections in CAR-T cell patients are mainly observed within 30 days after the infusion and are attributed to bacterial and viral species [14,15,16]. Additionally, invasive fungal disease (IFD) can be observed in these patients, leading to increased mortality and morbidity [17,18]. Data regarding epidemiology, risk factors, management, and outcomes of IFDs in CAR-T cell recipients are scarce. Moreover, a lack of large real-world studies about the incidence of IFDs in these patients has been recognized.

In the current literature review, we searched the recent literature, and the published data regarding the pathogenesis-risk factors, diagnosis, epidemiology, management, and outcomes of IFDs in pediatric and adult patients who receive CAR-T cell immunotherapy are summarized. Moreover, the role of antifungal prophylaxis in these patients is examined. In the era of precision medicine and novel therapeutics, more data regarding IFDs in the post-CAR-T cell infusion period are essential. 

## 2. Pathogenesis-Risk Factors

### 2.1. Pre-Infusion Factors: Underlying Hematological Malignancies and Previous Treatments

Various factors, including diagnosis of B-ALL, history of allogeneic HCT (allo-HCT), history of IFD prior to CAR-T cell therapy, and increased number of previous treatment lines, have been recognized as infection risk factors in CAR-T cell recipients [19,20,21,22,23,24]. Specifically, previous allo-HCT is a well-established risk factor for IFD development, which is more common in patients with R/R B-ALL in comparison to those with lymphomas or MM [25,26]. In patients who receive CAR-T cell immunotherapy, and who have also undergone allo-HCT, a history of graft-versus-host disease diagnosis results in increased risk for IFDs [27]. Furthermore, the administration of targeted therapeutic agents, such as ibrutinib, rituximab (B cell-depleting agent), and alemtuzumab, often leading to impaired cell mediated immunity, before CAR-T cell therapy might increase the IFD risk in these patients [20,28,29]. More real-world data are of paramount importance in understanding the role of these agents in IFD development post-CAR-T cell infusion. In addition to the above, patients’ comorbidities might increase the risk for IFDs. For example, uncontrolled diabetes mellitus has been recognized as a risk factor for mucormycosis [30,31,32].

### 2.2. CRS and ICANS

CRS and ICANS are the main toxicities developed in the post-infusion period [33]. These complications and the use of corticosteroids for their management are included in the risk factors for infectious complications in CAR-T cell recipients. Interestingly, CRS grade has been associated with the severity of infections in the post-infusion period [34]. Endothelial injury, immune dysregulation, and mucositis are implicated in CRS pathogenesis, which may lead to IFD susceptibility [35]. In addition to corticosteroids, other immunosuppressive agents, such as tocilizumab, and invasive measures (mechanical ventilation, central venous catheterization, hospitalization in intensive care unit [ICU]) are included in the management of these toxicities, increasing the risk for IFDs [36]. High doses of corticosteroids used for the treatment of severe CRS forms, such as hemophagocytic lymphohistiocytosis, have been recognized as a risk for factor IFD development in these patients [32,37,38]. Data regarding the association of IFDs and tocilizumab in CAR-T cell recipients are lacking; however, tocilizumab use in patients with severe coronavirus disease 2019 (COVID-19) infection has been associated with increased prevalence of fungal infections [39,40,41,42]. Future research on the impact of immunosuppressive agents used for CRS and ICANS treatment in the incidence of IFDs is essential.

### 2.3. Hematological Toxicity: Neutropenia

Severe neutropenia (absolute neutrophil count (ANC) < 500 cells/μL) is frequently observed in CAR-T patients after the administration of lymphodepleting chemotherapy with median duration below 10 days [43,44,45]. CRS and systematic inflammation are implicated in CAR-T-related neutropenia development [46]. Biphasic neutropenia is also common in these patients: within 3 weeks post-infusion is observed an intermittent recovery in ANC (phase I), while in 2 months, approximately, a second decrease is evident (phase II) [47]. CRS and systematic inflammation are implicated in CAR-T-related phase I neutropenia development [46]. The second neutropenic phase is attributed to suppression of bone marrow neutrophil production, and immune-mediated mechanisms have an important role [48]. Prolonged and severe neutropenia are well-established drivers of IFDs in patients with hematological malignancies [49,50]. Cases of IFDs have been described in patients with refractory and severe neutropenia post-CAR-T cell infusion [33,51].

### 2.4. Hematological Toxicity Beyond Neutropenia

CAR-T-related prolonged hematotoxicity is the most common late-onset adverse event (after the first month of CAR-T infusion) in real-world clinical settings [52]. Hematotoxicity leads to severe infections, hemorrhagic events, longer hospitalization, and more frequent follow-ups [53]. Beyond neutropenia, CD4^+^ T cell-mediated immunity has been described as impaired in these patients [54]. Persisting lymphopenia has been recognized as a risk factor for late-onset (>3 months post-infusion) IFD development and mainly for *Pneumocystis jirovecii* pneumonia (PJP). In the study of Little et al., 280 CAR-T cell recipients were studied and PJP was observed in 3 patients with lymphopenia. The median time of PJP onset was 390 days (range: 115–441) [55]. Moreover, PJP infections in the cohort of Baird et al. occurred in patients with CD4^+^ T cell count below 200 cells/μL [56]. B cell aplasia, hypogammaglobinemia, and antibody deficiency, observed in these patients, might lead to increased IFD risk due to their indirect impact on T cell-mediated immunity [56,57]. Zahid et al. described a case of disseminated coccidioidomycosis after CAR-T cell immunotherapy, which was attributed, among other factors, to an IgG subclass deficiency, as a result of B cell depletion [58]. Predictive score systems, incorporating pre- and post-infusion variables, should be developed to predict IFD onset in these patients [59]. In Figure 1, the risk factors implicated in IFD pathogenesis in CAR-T cell patients are summarized.

## 3. Diagnosis of IFD in CAR-T Cell Setting

Although major improvements have been made in the diagnosis of IFDs, their differentiation from non-fungal infections and non-infectious complications, such as CRS, still remains challenging and often eludes clinical suspicion. The cornerstones of diagnosis are histopathological examination of infected tissue, chest computed tomography (CT) scans, and microbiological tests, both culture-based and non-culture-based, with serological testing, polymerase chain reaction (PCR), and mass spectrometry being steadily integrated into laboratory protocols [60]. Diagnosis can be classified as proven, probable, and possible, based on the revised and updated version of the consensus definitions of IFDs published by EORTC/MSGERC [61]. Proven diagnosis requires the detection of pathogenic fungi through histopathological or culture methods from sterile sites. For a probable diagnosis, all three of the following criteria must be met: (i) host factors (immunosuppression, neutropenia, etc.), (ii) relevant clinical signs and symptoms, and (iii) mycological evidence, such as positive imaging or molecular testing. A possible diagnosis requires two of the three aforementioned criteria to be present [61]. Histopathological examination is the most certain way to establish a proven diagnosis; however, the invasiveness of biopsy should always be taken into consideration, especially for recipients of CAR-T cell therapy who are immunocompromised and might also suffer from insufficient blood coagulation.

The methods of choice for the identification of fungal pathogens are culture and direct microscopy of biological fluids including blood serum, bronchoalveolar lavage (BAL), sputum, cerebrospinal fluid (CSF), and others, owning to their minimal risk to patient safety and proven specificity. Additionally, fungal culture allows for the determination of antifungal susceptibility, as well as the identification of fungal species. Culture remains the gold standard diagnostic method for invasive candidiasis. However, it also presents some considerable drawbacks, such as the long turn-around time (2 to 5 days for yeast and yeast-like fungi and 7 to 10 days for filamentous fungi), possible contamination with irrelevant microorganisms, and low sensitivity values [62].

As for serological testing, major advancements have paved the way for the inclusion of practices such as the *Aspergillus* galactomannan (GM) antigen test and the (1,3)-β-d-glucan (BDG) test in laboratory routine. The GM antigen test is measured by a sandwich enzyme immunoassay which detects galactomannan, a polysaccharide cell wall component that is mainly released by *Aspergillus* during growth in infected blood, bronchial, and other body fluids [63]. It has been proven useful in the diagnosis of aspergillosis in CAR-T cell therapy recipients, and its values have been associated with clinical outcomes [19]. However, it cannot give a proven diagnosis, since false positive and false negative results are not uncommon. False positive results have been associated with MM and the intake of semi-synthetic β-lactam antibiotics and probiotic supplements, all of which are common in CAR-T cell immunotherapy recipients. False negative results have been associated with concomitant use of mold-active antifungal agents [60]. Therefore, the GM antigen test should be used in conjunction with other microbiological methods, notably culture and/or PCR.

The same is true for the BDG test, which detects a cell wall polysaccharide, (1-3)-β-d-glucan, in patient serum, a component present in most fungi, including *Candida* spp., *Fusarium* spp., and *P. jirovecii*. It is notably absent from cryptococci, the *Mucormycetes*, and *Blastomyces dermatitidis* [64]. It has been proven as a useful tool in the diagnosis of IFD and fungemia [65]. Similar to the *Aspergillus* galactomannan antigen test, a tendency for false positive results has been noticed in patients receiving semi-synthetic β-lactam antibiotics and those suffering from bacterial bloodstream infections (e.g., *Pseudomonas aeruginosa*), while false negative results have been observed in patients under concomitant use of antifungal agents. CAR-T cell therapy recipients who receive multiple prophylactic antifungal agents (fluconazole, posaconazole, and oral trimethoprim/sulfamethoxazole) could belong to the aforementioned categories. Furthermore, we have to underline that the use of BDG detection in pediatric patients is limited due to a high proportion of false positive results. An exception could be PJP with out-of-scale BDG in the presence of the right clinical condition, but this needs further validation [66].

PJP is a rare but documented infectious complication of CAR-T cell therapy [13]. It mostly occurs in later stages following the infusion (from 50 up to 250 days later) and is linked to the observable long-term B cell aplasia and subsequent hypogammaglobulinemia. Since it can manifest so long after the infusion, it is often underappreciated, but nevertheless life threatening. In the case of clinical suspicion of PJP, the recommended method for identification is immunofluorescent microscopy or molecular techniques, which have been found to exhibit higher sensitivity [67].

Apart from culture, microscopy, serological tests, and histopathological examination, three more methods have been proven useful for rapid fungal identification. Those include imaging techniques, PCR, and mass spectrometry. Regarding imaging techniques, the preferred method for fungal identification is high-resolution computed scanning. Nodules or infiltrates with or without a halo sign, found in CT scans, remain a useful and rapid find for the identification of invasive pulmonary aspergillosis (IPA), especially in neutropenic patients, such as those receiving CAR-T cell immunotherapy [68]. Following neutrophil recovery, an air crescent sign may develop, though it remains a nonspecific and rather late find [61].

PCR-based methods have also been developed. For a long time, they lacked standardization and there were only a few commercially available PCR assays. However, over the past decade, major progress has been made in standardizing the available protocols by the European *Aspergillus* PCR Initiative (EAPCRI) [69]. With extensive evaluation of the assays, the specificity and sensitivity for plasma *Aspergillus* PCR reached 83.3% and 94.7%, respectively. Those results make the PCR-based methods comparable to the more widespread GM antigen and BDG tests, and they can be used to verify previous results.

Finally, the latest innovation to find application in mycological diagnostic procedures is matrix-assisted laser desorption/ionization time-of-flight mass spectrometry (MALDI-TOF MS) [70]. This novel technology was launched around 2010 and has lately found great success in the identification of filamentous and yeast-like fungi, utilizing raw samples taken directly from the culture or blood cultures [71]. Although it has yet to find mainstream use due to its initial high acquisition cost and the newly founded fungal databases, many studies indicate its effectiveness in the identification of cryptic and newly emerging fungal species which often tend to display higher minimum inhibitory concentrations (MIC) to antifungal agents. Such strains are often opportunistic or nosocomial and therefore are more dangerous to immunocompromised patients, such as hematological patients. MALDI-TOF MS has the potential to revolutionize phenotypical fungal identification, but until the databases provided with the equipment are up to date with the increasing number of emerging fungal species, it cannot be solely trusted for a proven diagnosis. Therefore, it is advised that it must be used in combination with other identification techniques [72].

## 4. Epidemiology of IFDs Post-CAR-T Immunotherapy

### 4.1. Epidemiology of IFDs in Adult Patients

The prevalence of IFDs in adult CAR-T product recipients varies between studies from 0–12%. In several patient cohorts, no IFDs have been identified. In a study of 26 individuals with R/R MM, no fungal infection was developed, with a median follow-up period of 8.9 (3.1–18.1) months; in this patient cohort, 9 out of 26 (34.5%) patients developed CRS, while anti-mold and anti-yeast prophylaxis was administered as well [73]. Additionally, zero IFDs are reported in the first 30 days post-CAR-T therapy in 72 individuals with ALL and non Hodgkin lymphoma (NHL) [22]. No IFDs were reported, moreover, in a cohort of 41 patients with DLBCL (treated with agenlecleucel and oraxicabtagene ciloleucel) in the course of a 30-day follow-up period [74]. In the study of Wittmann Dayagi et al., the prevalence of IFDs was 1.5% and notably no antifungal prophylaxis was administered [75]. Two phase 2 clinical trials, CARTITUDE-1 and CRB-401, did not report any IFDs nor any fungal disease. CARTITUDE-1 was a multicenter trial (16 centers in the USA) with 97 MM patients between July 2018 and October 2019 [76]. CRB-401 was conducted in 33 patients with R/R MM between 2016 and April 2018 [77]. A low prevalence of IFDs was reported in TRANSCEND, a multicenter, multicohort, prospective study at 14 cancer centers in the USA [78]. In a study developed by our group, none of the 19 CAR-T cell recipients developed an IFD [79].

In a single-center retrospective study by Mikkilineni and colleagues, the prevalence of IFDs was 3.3% (2 of 162 patients), and fatal events were reported in a follow-up period of one month [22]. In a different cohort of 99 patients, 3 developed IFDs (2 invasive pulmonary aspergillosis and 1 proven *Candida* peritonitis) between the first and 100th day after CAR-T cell therapy, while no fungal infections developed after 100 days of follow-up [80]. Additionally, 1 case of fatal aspergillosis (in a patient with hemophagocytic lymphohistiocytosis) was reported among 99 individuals. Data from the JULIET trial showed that 167 patients with DLBCL presented 3 IFDs (2 *Candida* and 1 *Aspergillus*), but no data on the clinical outcomes of the infections are available [81]. In the Zuma-1 study (a multicenter trial with patients from the USA and Israel), 108 individuals with DLBCL received axicabtagene ciloleucel [82]. Among those 108, 3 developed IFDs (2 *Candida* and 1 *Aspergillus*).

IFD prevalence was 6% (n = 2) in a cohort of 32 patients with R/R MM, while in this study an increased cumulative incidence of IFDs was reported during days 150–180 after the CAR-T cell therapy [83]. Similar data are presented in the results of two retrospective studies of 52 and 55 individuals with R/R MM (prevalence 5.7% and 6%, respectively) [84,85]. Additionally, in the study by Kambhampati et al., the IFDs were observed 120–150 days after CAR-T cell therapy, and the majority of IFDs identified were attributed to *Aspergillus fumigatus* (in 2 of 3 patients) [85].

The prevalence of IFDs in patient cohorts with ALL varies. In the study of Wang et al., 3 among 76 CAR-T cell recipients developed IFDs [44]. Vora and colleagues conducted a study on 83 young adults (aged ≥ 18 and < 25 years old) and children [24]. In this cohort, 1 IFD (proven pulmonary infection with *Cunninghamella*) was reported on day 21 post-therapy. In a single-center retrospective study by Czapka et al., 5 among 73 patients with predominantly NHL (97%) developed an IFD, 4 attributed to invasive yeast infections (*Candida* species) and 1 to an invasive mold infection (pulmonary aspergillosis) [86]. The prevalence was higher in the first 30 days after CAR-T cell therapy, and it was zero one year after the therapy, in a multicenter retrospective analysis of 248 patients with NHL who received CAR-T cell therapies; 24 developed IFDs, and 3 of these patients deceased. The IFDs were predominantly present in patients with high CAR-HEMATOTOX (HT) scores [47]. Baird and colleagues investigated the development of infective complications in 41 patients with DLBCL who received axicabtagene ciloleucel therapy [56]. In this single-center study, 6 IFDs (mostly from *Candida* species) developed, and the cumulative incidence of these was calculated to be 8.9% in the follow-up period (12 months). The retrospective study developed by Zhu et al. reported a similar prevalence (8.7%) of IFDs in a population of 92 individuals, but in this cohort of patients, the majority were observed during the first 180 days after the CAR-T cellular therapy [87]. In specific patient cohorts, the mortality in patients with IFDs is high (up to 50%), while the majority of the deaths are observed in patients with rapid disease progression [88]. Not all IFDs are presented in the initial 180 days after the CAR-T therapy. For instance, in a patient cohort at Memorial Sloan Kettering Cancer Center the 1-year cumulative incidence was 4.0 (95% CI, 7.0–12.3) vs. 1.7 (95% CI, 0.1–7.9) at 3 months of follow-up [89].

In a population of 39 adults and children (mean age 52) with R/R ALL or NHL, two presented IFDS, one patient presented *Candida* fungemia, and one patient presented a skin and soft tissue fungal infection due to *Aspergillus* and *Rhizopus* [90]. Hill et al. reported 6 IFDs in 4 adults post-CAR-T therapy; all individuals who presented an IFD had severe grade CRS (≥3) [21]. In this study, the IFDs predominantly occurred in the first 28 days after the CAR-T cell therapy (risk ratio (RR) 0.56; 95% confidence interval (CI), 0.33–0.93; P 5.02). Garner et al. reported a 10% prevalence of fungal infections among 103 patients; additionally, 2 out of 10 patients with an IFD deceased due to the infection [51]. In a different cohort of 280 adult patients with NHL, 8 out of 280 developed an IFD [55]. A fact that should not be disregarded is that patients in this retrospective study did not receive any antifungal prophylaxis. Data from NCT04531046 report 2 fatal cases of IFDs: 1 pulmonary mucormycosis in a 74-year-old male with grade 4 CRS and grade 4 ICANS (63 days post-infusion), and 1 bronchopulmonary aspergillosis in a 66-year-old male with grade 1 CRS and grade 3 ICANS (107 post-infusion) [91]. Park et al. reported a 7.5% prevalence of IFDs in a population of 53 individuals with B-ALL, and the onset of the infection was at 23 days post-CAR-T injection (interquartile range (IQR) 20–29 days) [15]. In the retrospective study of Wudhikarn et al., in which 60 patients who underwent CAR-T cell therapy were enrolled, 2 cases of IFDs were reported, with 1-year cumulative incidence of 4.0 (95% CI: 7.0–12.3) [92]. A study by Munshi of 128 patients with MM who received idecabtagene vicleucel reported 11 fungal infections [93]. In a retrospective observational study conducted by Cordeiro et al., the majority of IFDs were late infections (persisted and/or occurred ≥90 days after the therapy) and involved *Aspergillus* (n = 2), *Candida* (n = 1), and *Coccidioides* (n = 1) species [94].

A case series of Haidar and colleagues reported two cases of IFDs [33]. The first patient was a 23-year-old woman with refractory ALL, and she presented with left maxillary invasive fungal sinusitis caused by *Fusarium solani*. Liposomal amphotericin B, voriconazole, terbinafine, and amphotericin B nasal irrigations were administered, and surgical management was implemented in this patient’s case. The second was a 56-year-old patient with hairy cell leukemia (HCL) left orbital cellulitis and left sphenoid invasive fungal sinusitis caused by invasive mucormycosis with negative cultures; the patient received liposomal amphotericin B, caspofungin, and posaconazole and underwent surgical therapy for the IFD as well. The patient deceased due to the relapse of the HCL. Cheok et al. presented a case of invasive mucormycosis in a 54-year-old patient with relapsed primary central nervous system lymphoma and steroid-induced diabetes [32]. *Rhizopus* microspores were identified on the microscopy of surgical specimens; the patient received dual antifungal therapy (liposomal amphotericin-B and intravenous isavuconazole), underwent surgical operations, and survived. The findings of the above studies are summarized in Table 1.

*Aspergillus* spp., *Candida* spp., and *Pneumocystis jirovecii* are the cause of almost 90% of IFDs following CAR-T cell therapy, as indicated by large scale database analysis conducted by Sassine J. et al., where 256 patients from a sample of 2256 (11.4%) presented a fungal infection [95]. Amongst those were 175 with a *Candida* infection, 46 with an *Aspergillus* infection, and 28 with a *Pneumocystis* infection. The rest (23) had other infections, caused by *Histoplasma* (16), *Mucormycetes* (5), and *Cryptococcus* (2).

### 4.2. Epidemiology of IFDs in Pediatric Patients

IFDs have also been recognized as an important infectious complication in pediatric patients who receive CAR-T cell immunotherapy, with a notable impact on morbidity and mortality [96,97]. The risk factors for IFDs in these patients are similar to those in adult populations. Specifically, previous treatment lines of chemotherapy or radiotherapy for the underlying malignancy lead to immunosuppression [98,99]. Especially in children who receive targeted therapies, possible IFDs require individualized clinical monitoring, along with frequent laboratory testing and antifungal prophylaxis in selected cases [29,100]. As confirmed in most studies in pediatric settings, fungal pathogens that cause IFDs can vary. In the most common pathogen species, *Candida* spp., causing candidemia or invasive candidiasis infections (ICIs) (yeast infections), and *Aspergillus* spp., causing invasive aspergillosis infections (IAIs) (mold infections), are included [100,101]. ICIs are more frequently reported in pediatric ICUs, while IAIs are mostly observed in children with underlying malignancies [101].

Few studies examining the prevalence and risk factors of IFDs following CAR-T cell infusion in pediatric patients alone have been published. In the clinical trial of Ghorashian et al., 14 patients with ALL were included, and 2 cases of chest IFDs were described [102]. In the aforementioned study of Vora et al., including both pediatric and young adult patients, 1 case of IFD was reported among 83 CAR-T cell recipients with a median age of 12 years (range: 1–25) [24]. In the clinical trial of Maude and colleagues, 75 children were treated with tisagenlecleucel for B-ALL, and a case of fatal mucormycosis was described [103]. In a retrospective study of 79 pediatric patients with B-ALL who received CAR-T cell therapy, 1 fungal infection case was reported (a urinary tract infection caused by *Trichosporon asahii*) [104]. Maron et al. described two IFD cases among 38 CAR-T cell patients with a median age of 9.06 years (range: 1.8–23.6) [105]. In this study, antifungal prophylaxis with intravenous micafungin, followed by oral voriconazole until neutrophil recovery (ANC  ≥  500 cells/µL for at least 3 consecutive measurements), was given. Diamond et al., in their study, reported 1 case of probable pulmonary aspergillosis out of 27 children who underwent CAR-T cell immunotherapy for B-ALL [106]. Future studies examining the epidemiology of IFDs in pediatric CAR-T patients are of paramount importance, given the lack of data in this population.

The timing of IFDs can vary, given that invasive yeast infections, such as candidiasis, present in the first 30 days after CAR-T therapy, while invasive mold infections can also develop after this period [13]. For the diagnosis of such infections, the classification of proven, probable, and possible cases, as in adult patients, can be used [61]. Nowadays, novel diagnostic tools, such as molecular testing and fungal serum biomarkers, have been found helpful in the diagnosis of IFD cases [99]. The high rates of IFD mortality underline the need for early detection, prompt diagnosis, and immediate management of infections with antifungal agents [99].

## 5. Antifungal Prophylaxis in CAR-T Cell Recipients

Antifungal prophylaxis, as a strategy, aims to avoid further adverse effects, delays, and/or complications of the underlying disease [107]. Antifungal prophylaxis constitutes a widely used option during and after CAR-T immunotherapy to mitigate the high morbidity and mortality rates that are associated with IFDs, especially in patients with prolonged neutropenia [13,108]. The choice of prophylaxis is still debated, and so antifungals are frequently recommended to be administered after the therapy, depending on the patient’s condition, individualized for each one [33,57,109].

In the determination of this prophylactic strategy, several factors are assessed before the final selection of certain agents. Their safety and efficacy are evaluated and taken into account, always individualized for each patient [33]. Azoles are the antifungal agents with the widest usage both in adult and pediatric populations [60]. Fluconazole represents the first option against yeast infections, like those caused by *Candida* species, and is administered during periods of neutropenia, while voriconazole and posaconazole are recommended to prevent *Aspergillus* species [109]. Nowadays, isavuconazole is also suggested as a possible choice and, compared to others, provides several advantages [110]. The adjustment of the dosages is based on the condition of each patient, along with the body weight, mostly in children. Echinocandins are the drugs of second choice for patients unable to tolerate azoles, along with liposomal amphotericin B (LAMB) [111].

Recently published data in patients with hematological malignancies recommended the use of prophylaxis in periods of neutropenia and especially long-lasting neutropenia, independent of the underlying disease [109,112]. However, patients like CAR-T recipients, with a shorter expected duration of neutropenia (ANC < 500 cells/μL for ≤7 days), are not at increased risk of a fungal infection and should not be classified under prophylaxis [109]. In 2022, Little et al. presented their data regarding 280 patients who did not receive prophylaxis after CAR-T cell immunotherapy, and only 8 cases of fungal infections (2.9% of the population) were reported [55]. Of those, 5 patients developed early IFDs (3 IMIs, 2 IYIs) and 3 patients developed late infections with PJP; 4 of these patients died (1.4% of the study population) [55]. In the review of Kampouri et al., published in 2023, the existing guidelines for infection prevention in CAR-T cell therapy recipients were summarized [113]. As antifungal prophylaxis, the European Society for Blood and Marrow Transplantation (EBTM), the European Haematology Association (EHA), and the Société de Greffe de Moelle et de Thérapie cellulaire (SFGM-TC) recommend fluconazole, posaconazole, or micafungin if severe or prolonged neutropenia has a duration of more than 14 days, and/or long-term or high-dose (>3 days) steroids, in post-allo-HCT [113]. Fred Hutch (United States), CHUV Lausanne (Switzerland), and the Spanish group (Spain) suggest the use of fluconazole during neutropenia, while Dana Farber (United States) and LMU Munich (Germany) are against antifungal prophylaxis [113]. Regarding prophylaxis against PJP, all current guidelines from international societies mentioned above consider the administration of trimethoprim–sulfamethoxazole (TMP-SMX) twice daily for 3 days per week or a single dose every day. PJP prophylaxis has to be initiated at lymphodepleting chemotherapy and continue for 1 year until CD4 > 200 cells/mm^3^ [13,113].

Guidelines about antifungal exposure during CAR-T therapy exist mainly for adult patients since pediatric research is more limited and just characterizes the appropriate use of prophylaxis against viral, bacterial, and fungal pathogens [114]. As a general principle, antifungal prophylaxis is recommended in high-risk patients, but usually pediatric cases follow the guidelines that already exist for adults with several risks, given that there is no specific proof about the safe use of antifungal prophylaxis on this population. Further research into pediatric CAR-T recipients, with a focused interest in the role of antifungal prophylaxis, is needed. In general, for both adult and pediatric settings, research should emphasize individualized recommendations after an initial classification of the patients, based on their epidemiological characteristics, their condition, their risk factors, and treatment endpoints [13,33,57].

## 6. Management of IFDs in CAR-T Cell Recipients

According to the 8th edition of the EBMT Handbook for Haematopoietic Cell Transplantation and Cellular Therapies, three main strategies are considered when treating an IFD [60]. The first among those is referred to as empirical antifungal therapy and consists of administering broad-spectrum antifungals to neutropenic patients who experience fever for 5 to 7 days after adequate antibacterial coverage. Although this practice is not entirely supported by scientific evidence and most likely leads to increased toxicity and costs, it is favored by centers with limited access to mycological and radiological diagnostic tools. More specifically, the European Conference on Infections in Leukemia (ECIL) recommends the use of caspofungin (50 mg/day following 70 mg on day 1) or liposomal amphotericin B (3 mg/kg).

The second approach, the diagnostic-driven or pre-emptive approach, aims to begin antifungal therapy in high-risk patients upon early positive markers of fungal infection, such as a positive PCR, GM antigen, or BDG test and even indications during CT scanning [115]. A recent study by Maertens et al. illustrates the differences between empirical and diagnostic-driven approaches [116]. The results show that the overall survival for either one of the groups at day 42 and day 84 showed no significant statistical difference. However, there was a significant reduction in the usage of antifungals among the participants of the diagnostic-driven therapy, highlighting its financial benefits over the empirical approach. Lastly, the third approach relies upon the proven or probable diagnosis of IFD and is tailored towards each respective fungi identified.

Regarding IPA, the Infectious Diseases Society of America strongly recommends primary treatment with voriconazole, while equally effective alternative therapies include isavuconazole and liposomal amphotericin B (AmB), see Table 2 [117]. Those should be considered in the case of voriconazole intolerance, drug interactions (such as cyclosporine, tacrolimus, and sirolimus, and other CYP3A4 substrates such as tyrosine kinase inhibitors), prior exposure to broad-spectrum azoles (e.g., prophylaxis), or documented azole resistance. As of the last few years, isavuconazole is considered an equally effective alternative to voriconazole, following the 2017 ECIL recommendations, since it presents similar performance while exhibiting better tolerance and fewer drug interactions [118]. The use of echinocandins and the combination of antifungals as first-line treatment are not recommended. Additionally, the elimination of immunosuppressants (if possible) is recommended in anti-aspergillus therapy, as well as the reduction of corticosteroids, since azoles have been found to increase their levels. Other treatment methods include surgical debridement in easily accessible areas (e.g., in invasive fungal sinusitis or localized cutaneous disease) and granulocyte transfusions for severely neutropenic patients or IPA unlikely to respond to standard therapy. Lastly, therapy should last for at least 6 to 12 months and continue long after the dissipation of initial symptoms. According to the ECIL-6 guidelines, therapeutic drug monitoring for voriconazole (plasma target 1–6 mg/L), isavuconazole (target 2–4 mg/L), and posaconazole (plasma target > 1 mg/L) is strongly recommended, while the testing of *Aspergillus* GM antigen is a helpful indicator of response to therapy [118].

ICIs and/or candidemia are primarily treated using echinocandins (caspofungin, micafungin, or anidulafungin), as shown in Table 2 [60,118]. Fluconazole is an alternative initial treatment in patients who are not critically ill when fluconazole resistance is considered unlikely. Other treatments include voriconazole and amphotericin B formulations. Following species identification and antifungal susceptibility testing, a step-down approach is recommended and treatment should generally continue for at least two weeks after clearance of *Candida* and resolution of symptoms. Intravenous catheter removal can be considered for neutropenic patients.

For patients with documented PJP, high doses of trimethoprim/sulfamethoxazole are recommended (Table 2) [119]. The combination of primaquine plus clindamycin is the preferred alternative for patients with documented intolerance to the recommended regimen. Treatment success should be first evaluated after 1 week, and in case of clinical non-response, pulmonary CT scan and bronchoalveolar lavage should be repeated to look for secondary or co-infections. Treatment duration typically is 3 weeks and secondary anti-PCP prophylaxis is indicated in all patients thereafter.

Finally, although mucormycosis is rare in CAR-T cell recipients, it is an infection characterized by its highly destructive nature and rapid development, and therefore treatment should begin immediately upon the confirmation of the disease, following the initial symptoms. The guidelines of the European Confederation of Medical Mycology (ECMM) published in 2019 indicate that surgical debridement with clean margins should be initiated immediately, even if proven destructive to surrounding tissue [120]. Following the debridement, drug treatment should also begin without delay with daily doses of liposomal amphotericin B ranging from 5 mg/kg per day to 10 mg/kg per day. Recipients of increased doses tend to have increased response rates. Alternatives include isavuconazole and posaconazole but should be mostly conserved for salvage treatment. The duration of the therapy has high variability ranging from weeks to months and, in the meantime, antifungal susceptibility testing is recommended to improve the dosage.

## 7. Conclusions

In the current review the epidemiology of IFDs in CAR-T cell recipients was investigated. IFIDs are regarded as the third most prevalent factor leading to post-immunotherapy infections, with bacterial and viral infections being first and second, respectively. Although bacterial and viral pathogens are much more common than fungi, IFDs are associated with increased mortality and morbidity, especially in cases when diagnosis and treatment are delayed. The prevalence of IFDs has been reported to be low (0–10%) in both pediatric and adult populations. Studies should use the diagnostic criteria of EORTC/MSGERC for the diagnosis and classification of IFDs, to better understand the actual prevalence of these complications. Moreover, more real-world data regarding the role of antifungal prophylaxis in these patients are of paramount importance. Future work should also focus on the estimation of the frequency and cumulative risk of IFDs after CAR-T cell immunotherapy based on the results of published studies.

## Figures and Tables

**Figure 1 pathogens-14-00170-f001:**
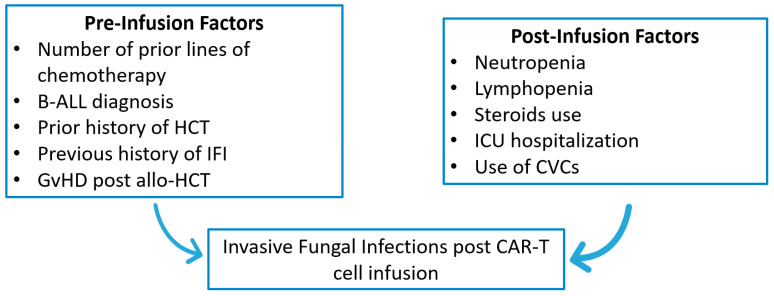
Risk factors implicated in IFD pathogenesis in CAR-T cell recipients. IFD: invasive fungal disease, CAR-T: chimeric antigen receptor-T, B-ALL: b-acute lymphoblastic leukemia, HCT: hematopoietic cell transplantation, GvHD: graft-versus-host disease, allo-HCT: allogeneic hematopoietic cell transplantation; ICU: intensive care unit, CVC: central venous catheter.

**Table 1 pathogens-14-00170-t001:** Studies examining the epidemiology of invasive fungal disease in CAR-T cell recipients.

Reference	Number of Patients (CAR-T Recipients)	Study Design	Study Period	Malignancy, Age, Median (Min–Max)	CRS, ICANS, N (%)	Prevalence of IFDs, N (%)	Mortality, N (%)	Follow-Up Period	Prophylaxis Anti-Yeast Anti-Mold
Gavriilaki et al., 2023 [79]	19	Retrospective study	2013 to 2022	ALL NHL NR	NR NR	0 (0)	Not develpoed IFDs	NR	NR
Mohan et al., 2021 [73]	26	Retrospective study	January 2019 to June 2021	R/R MM 65.5 (63–70)	9 (34.5) N/R	0 (0)	0(0)	8.9 (3.1–18.1) months	Yes Fluconazole Mold-active azole accompanied with steroids
Josyula et al., 2022 [83]	32	Retrospective study	January 2018 to February 2020	R/R MM 64 (44–77)	27 (84.4) 10 (31.3)	2 (6)	1 (33.3)	180 days	Patients with severe neutropenia received fluconazole
Tran et al., 2020 [90]	39 (Adult + peadiatric)	Retrospective study	NR	ALL NHL 52 (±22)	33 (85) NR	2 (5.2)	NR	180 days	Yes
Baird et al., 2021 [56]	41	Retrospective study	September 2017 to March 2019	DLBCL 56 (21–76)	Severe CRS (2.4) Severe ICANS (24.4)	6 (14.6)	0 (0)	Minimum 1 year	If patient has severe mucositis Fluconazole In known history of IMI posaconazole
Logue et al., 2022 [84]	52	Multicenter retrospective study	May 2021 to December 2021	R/R MM 66 (43–78)	44 (84.7) 10 (19.2)	3 (6)	0 (0)	90 days	Fluconazole Mold-active azole with steroids
Park et al., 2018 [15]	53	Phase I clinical trial	May 2010 to August 2016	R/R B-ALL 45 (IQR, 30–74)	45 (84.9) NR	4 (7.5)	1 (25)	As long as MSKCC lasted	Micagungin
Kambhampati et al., 2022 [85]	55	Retrospective study	2018 to 2020	R/R MM 62 (33–77)	48 (87) 8 (15)	3 (6%)	2 (66%)	Mean 6.0 months (95% CI 4.7–7.4)	In patients with ANC < 500/mm^3^
Beyar-Katz et al., 2022 [74]	60	Retrospective study	April 2019 to December 2020	DLBCL 69.3 (19.8–85.2)	44 (73) 16 (27)	0(0)	0(0)	30 days	Fluconazole at ANC < 500/mm^3^
Wudhikarn et al., 2020 [92]	60	Retrospective study	January 2018–June 2019	DLBCL 63 (19.5–85.9)	48 (80) 24 (40)	2 (4)	0(0)	1 year	Fluconazole If ANC < 500/mm^3^ Voriconazole
Garner et al., 2020 [51]	60	Retrospective study	July 2017 to May 2021	NHL 66 (23–84)	44 (73) NR	10 (10)	2 (20)	Minimum 6 months	NR
Houot et al., 2023 [91]	62	Open-label, phase 2 clinical trial	March 2021 to May 2022	DLBCL 70 (49–81)	58 (93.5) 32 (51.6)	NR	2	Median 12 months (2.1–17.9	NR
Mikkilineni et al., 2021, [22]	72	Retrospective study	2012 to 2018	ALL NHL 19 (4–69)	129 (79.6) 39 (24.1)	0 (0)	0 (0)	30 days	Micafungin Azole
Czapka et al., 2023 [86]	73	Retrospective study	Patients received CAR-T prior September 2021	NHL ALL 64 (51–75)	65 (89) 38 (52)	5 (6.8)	NR	Up to 2 years	Fluconazole or micafungin Mold-active azole with steroids
Wang et al., 2021, [44]	76	Retrospective study	April 2015 to September 2020	ALL 31.5 (15–74)	62 (81.6%) NR	3 (8)	NR	NR	NR
Vora et al., 2020 [24]	83	Retrospective study	2014 to 2017	ALL 12 (1–25)	74 (89) NR	1 (1.2)	0 (0)	90 days	Fluconazole Antimold azole Echinocandin
Logue et al., 2021 [88]	85	Retrospective study	February 2016 to February 2019	DLBCL 64 (28–79)	79 (92.4)	4 (4.7)	2 (50)	360 days	Fluconazole
Cordeiro et al., 2020 [94]	86	Retrospective study	July 2013 to February 2017	ALL NHL CLL 57 (23–75)	NR NR	4 (4.6)	NR	Median 34 months (18–44)	NR
Wittmann Dayagi et al., 2021 [75]	88	Retrospective study	July 2016 to May 2019	ALL NHL 33.36 (1.9–70.6)	65 (74) 43 (48)	1 (1.2)	0 (0)	60 days	No prophylaxis was administered
Zhu et al., 2021 [87]	92 (82 adults)	Retrospective study	July 2015 to May 2019	ALL NHL 35 (7–71)	61 (66.3) NR	8 (8.7)	NR	Up to 2 years	NR
Little et al., 2023 [80]	99	Retrospective study	November 2016 to May 2022	R/R MM 63 (34–78)	85 (85.9) 15(15.2)	3 (3)	1 (33.3)	1 year	Micafungin only for patients with >4 days of neutropenic fever
Locke et al., 2021 [82]	108	Μulticenter, open-label, single-arm, phase 1 and 2 trial	May 2015 to September 2016	DLBCL 56 (46–64)	95 (88) NR	3 (2.8)	NR	40·3 months (IQR 37.8–43.8)	NR
Hill et al., 2018 [21]	133	Phase 1/2 open-label clinical trial	Before September 2016	ALL CLL NHL 54 (20–73)	93 (69.9) 53 (39.8)	6 IFDS In 4 patients (3.0)	2 (33.3) ^1^	Up to 90 days	Fluconazole at ANC < 500/mm^3^
Mikkilineni et al., 2021 [22]	162	Single-center retrospective study	2012 to 2018	ALL NHL MM OS/NB 19 (4–69)	129 (79.6) N/R	2 (3.3)	0 (0)	30 days	Yes Micafungin Mold-active azole
Schuster et al., 2021 [81]	167	Μulticenter, open-label, single-arm, phase 2 trial	July 2015 to November 2017	DLBCL 56 (46–64)	66 (39.5) NR	3 (1.8)	NR	40·3 months (IQR 37.8–43.8)	NR
Rejeski et al., 2022 [47]	248	Multicenter retrospective study	May 2018 to June 2021	NHL 63 (19–83)	209 (90.6) 110 (44.4)	24 (96.7)	3 (12.5%)	90 days	Fluconazole
Abramson et al., 2020 [78]	269	Multicenter prospective study	January 2016 to July 2019	DLBCL 63 (54–70)	127 (47) NR	2 (1)	NR	Median 18.8 (15.0–19.3) months	NR
Little et al., 2022 [55]	280	Retrospective study	December 2017 to September 2021	NHL 64 (19–82)	239 (85) 153 (55)	8 (2.9)	1 (12.5)	Median 259 days (7–1340)	No

CAR-T: Chimeric antigen receptor-T, CRS: Cytokine release syndrome, ICANS: Immune effector cell associated neurotoxicity syndrome, IFD: Invasive fungal infections, R/R: Relapsed/refractory, MM: Multiple myeloma, N/R: Not referred, ALL: Acute lymphoblastic leukemia, NHL: Non-Hodgkin lymphoma, OS/NB: Osteosarcoma/Neuroblastoma, ANC: Absolute neutrophil count, DLBCL: Diffuse large B cell lymphoma, IQR: Interquartile range, CI: Confidence interval. All medians are presented as Median (min-max) and means as Mean (95% CI). ^1^ Infection was the primary cause of death in a patient with CLL who deceased of pulmonary hemorrhage due to an IFD caused by *Aspergillus ustus* 90 days post-CAR–T cell infusion. Infection was a contributing factor to death in a neutropenic patient with ALL who had fatal CRS and concurrent severe *Clostridium difficile* infection.

**Table 2 pathogens-14-00170-t002:** Antifungal agents for the management of IFDs in patients with hematological malignancies.

IFD	Antifungal Agent
IPA	Voriconazole, Isavuconazole, Liposomal Amphotericin B
ICIs and candidemia	Echinocandins, Fluconazole, Voriconazole, Amphotericin B
PJP	Trimethoprim/sulfamethoxazole, Primaquine plus clindamycin

IFD: invasive fungal disease, IPA: invasive pulmonary aspergillosis, ICIs: invasive *Candida* infections, PJP: *Pneumocystis jirovecii* pneumonia.

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
