# Peer review of "Invasive Fungal Disease After Chimeric Antigen Receptor-T Immunotherapy in Adult and Pediatric Patients"

_pathogens, 2025, doi:10.3390/pathogens14020170_

Round 1
Reviewer 1 Report
Comments and Suggestions for Authors
Evangelidis and colleagues from Greece submitted a review on IFI/IFD after Car-T therapy.
Comments:
In the abstract, the authors state that hypogammaglobulinemia is a risk factor for immunosuppression, which it is, but they don’t explain how it is a risk factor for IFI in the body of the paper. This concept needs more explanation since medical providers will want to know if IV IG repletion is important for patients who develop IFI. It is good that hypogammaglobulinemia does not appear in Figure 1.
- The authors report that they “searched systematically”, which is essentially saying they did a “systematic review”, but they don’t have a methods section. I think it is fine not to have a methods section, but then they should not use the term “searched systematically” because that has certain connotations for how they retrieved relevant publications. Please revise.
- The manuscript has variable ways they address genus and species names throughout. Latin genera and species should always be in italics. Genus should be capitalized, and species should not be capitalized.
- What does “in the study by colleagues” mean?
- What does “Not all published data clarify whether fungal infections were caused by IFDs” mean? Aren’t a fungal infection and an IFD the same thing?
- The sentence “A case series of Haidar and colleagues reported two cases of IFDs” needs to be referenced. Is it reference 33 or 51? To this reviewer, it appears to be the same reference but duplicated in the reference list.
- The sentence “Cheok et al. presented a case of invasive mucormycosis in a 54-year-old patient with relapsed primary central nervous system lymphoma and steroid-induced diabetes” is referenced to bibliographic entry #92, but the first author of #92 is Houot. Cheok is the first author of bibliographic entry #32, but is that a case report?
- Suggest adding an order to Table 1. Perhaps you could present the rows by the number of patients, starting with small numbers and going to larger numbers in each study represented?
- The first time that study bibliographic entry #95, Wudhikarn, appears is in Table 1. It should appear in the text, since every other study in Table 1 appears in the text.
- The sentence, “Antifungal prophylaxis, as a strategy, aims to avoid further adverse effects, delays, and/or complications of the underlying disease”, has a perfect reference with PMID 38481428, entitled: “Modeling Invasive Aspergillosis Risk for the Application of Prophylaxis Strategies”.
Author Response
Reviewer 1
Evangelidis and colleagues from Greece submitted a review on IFI/IFD after Car-T therapy.
Comments:
In the abstract, the authors state that hypogammaglobulinemia is a risk factor for immunosuppression, which it is, but they don’t explain how it is a risk factor for IFI in the body of the paper. This concept needs more explanation since medical providers will want to know if IV IG repletion is important for patients who develop IFI. It is good that hypogammaglobulinemia does not appear in Figure 1.
Response: We would like to thank the reviewer for the comments and time dedicated to reviewing our manuscript. Indeed the data we present about the role of hypogammaglobulinemia and the development of invasive fungal infections in these patients are confusing. In the revised version of our work, hypogammaglobulinemia was removed as a risk factor for these infections.
- The authors report that they “searched systematically”, which is essentially saying they did a “systematic review”, but they don’t have a methods section. I think it is fine not to have a methods section, but then they should not use the term “searched systematically” because that has certain connotations for how they retrieved relevant publications. Please revise.
Response: The reviewer is right. Our work is not a systematic review. We made the essential changes in the revised manuscript.
- The manuscript has variable ways they address genus and species names throughout. Latin genera and species should always be in italics. Genus should be capitalized, and species should not be capitalized.
Response: We are sorry for this error. The essential modifications have been made throughout the manuscript.
- What does “in the study by colleagues” mean?
Response: Thanks for this comment. Indeed, this point is not comprehensible. The phrase was replaced by “in the study by Kambhampati et al.”.
- What does “Not all published data clarify whether fungal infections were caused by IFDs” mean? Aren’t a fungal infection and an IFD the same thing?
Response: We are grateful for this comment. This sentence was removed in the revised version of the manuscript. Moreover, the following sentence: “The study by Nikhil and colleagues among 128 patients who received Idecabtagene Vicleucel with MM reports 11 fungal infections but is not clarified if the fungal infection provoked an IFD in these patients.” was converted to: “The study by Nikhil and colleagues among 128 patients who received Idecabtagene Vicleucel with MM reports 11 fungal infections”
- The sentence “A case series of Haidar and colleagues reported two cases of IFDs” needs to be referenced. Is it reference 33 or 51? To this reviewer, it appears to be the same reference but duplicated in the reference list.
Response: The reviewer is right. A reference was added to this point. Moreover, references 33 and 51 were duplicates.
- The sentence “Cheok et al. presented a case of invasive mucormycosis in a 54-year-old patient with relapsed primary central nervous system lymphoma and steroid-induced diabetes” is referenced to bibliographic entry #92, but the first author of #92 is Houot. Cheok is the first author of bibliographic entry #32, but is that a case report?
Response: We are grateful for this comment. In this sentence the reference by Cheok et al., in which a case report of mucormycosis along with system analysis and literature review are described, is now cited.
- Suggest adding an order to Table 1. Perhaps you could present the rows by the number of patients, starting with small numbers and going to larger numbers in each study represented?
Response: We are thankful for this interesting idea! We organized table 1 starting with studies with small numbers of patients and going to larger numbers, as you suggested.
- The first time that study bibliographic entry #95, Wudhikarn, appears is in Table 1. It should appear in the text, since every other study in Table 1 appears in the text.
Response: Thanks for this idea! The following sentence was added: “In the retrospective study of Wudhikarn et al., in which 60 patients who underwent CAR-T cell therapy were enrolled, 2 cases of IFDs were reported, with 1 year-cumulative incidence 4.0 (95% CI: 7.0–12.3).”
- The sentence, “Antifungal prophylaxis, as a strategy, aims to avoid further adverse effects, delays, and/or complications of the underlying disease”, has a perfect reference with PMID 38481428, entitled: “Modeling Invasive Aspergillosis Risk for the Application of Prophylaxis Strategies”.
Response: We are grateful for the suggestion of this interesting reference. It was incorporated into the novel version of the manuscript.
Reviewer 2 Report
Comments and Suggestions for Authors
The paper is useful since it summarize data on IFD after CAR-T therapy. However, I found some deficiencies and typo errors that must be corrected
1. There is no description on how "literature was systematically reviewed" (line 73). This should be clearly specified.
2. Since the paper regards also pediatric patients it should be stated that detection of BDG is considered useless (see also ECIL-8 recommendations) because of a high proportion of false-positive results. A possible exception could be PJP with out-of-scale BDG in presence of the right clinical condition, but this need further confirmations (J Fungi (Basel). 2024 Apr 9;10(4):276)
3. In many cases the name of the pathogens was written in a wrong way: they should be written in italica and the genus with the first letter in uppercase.
4. the name of Coccidioides was written in a wrong way (line 320)
5. As for diagnostic-driven therapy of IFD its effectiveness has been documented also in pediatrics and not only in adults (ref 114), and this should be added ( J Antimicrob Chemother. 2018 Oct 1;73(10):2860)
6. I found the figure useful bit poor (e.g. need to insert GvHD after allo HSCT). Needs to be improved
Finally, if it would be possible I would like to have an estimate of the frequency (episodes/months of follow up), and cumulative risk of IFD after CAR-T. These measures could be very useful for the design of strategies. If data are not available /retrivable this should be stated
Author Response
Reviewer 2
The paper is useful since it summarize data on IFD after CAR-T therapy. However, I found some deficiencies and typo errors that must be corrected
Response: We are grateful for the time that the reviewer dedicated to reviewing our manuscript. Indeed the reviewer’s comments were valuable for the quality of our work.
- There is no description on how "literature was systematically reviewed" (line 73). This should be clearly specified.
Response: Indeed the reviewer is right. This sentence was modified to: “In the current literature review, we searched systematically the recent literature, and the published data regarding the pathogenesis-risk factors, diagnosis, epidemiology, management, and outcomes of IFDs in pediatric and adult patients who receive CAR-T cell immunotherapy are summarized.”
- Since the paper regards also pediatric patients it should be stated that detection of BDG is considered useless (see also ECIL-8 recommendations) because of a high proportion of false-positive results. A possible exception could be PJP with out-of-scale BDG in presence of the right clinical condition, but this need further confirmations (J Fungi (Basel). 2024 Apr 9;10(4):276)
Response: Thanks for this comment. The following sentences were included in the novel version of the manuscript: “Furthermore, we have to underline the use of BDG detection in pediatric patients is limited due to a high proportion of false-positive results. An exception could be PJP with out-of-scale BDG in the presence of the right clinical condition, but this needs further validation”
- In many cases the name of the pathogens was written in a wrong way: they should be written in italica and the genus with the first letter in uppercase.
Response: The reviewer is right. The essential changes have been made throughout the manuscript.
- the name of Coccidioides was written in a wrong way (line 320)
Response: We are sorry for this error. We made the essential modification in the novel version of the manuscript.
- As for diagnostic-driven therapy of IFD its effectiveness has been documented also in pediatrics and not only in adults (ref 114), and this should be added ( J Antimicrob Chemother. 2018 Oct 1;73(10):2860)
Response: Thanks for this interesting idea! This reference was incorporated in the revised manuscript.
- I found the figure useful bit poor (e.g. need to insert GvHD after allo HSCT). Needs to be improved
Response: We are grateful for this suggestion. The role of GvHD after allo-HSCT was incorporated in Figure 1.
Finally, if it would be possible I would like to have an estimate of the frequency (episodes/months of follow up), and cumulative risk of IFD after CAR-T. These measures could be very useful for the design of strategies. If data are not available /retrivable this should be stated
Response: Thanks for this idea! This would be a very interesting future approach. The following phrase was added in the conclusions section: “Future works should also focus on the estimation of the frequency and cumulative risk of IFD after CAR-T cell immunotherapy based on the results of the published studies.”
Round 2
Reviewer 1 Report
Comments and Suggestions for Authors
Evangelidis and colleagues from Greece submit a revised manuscript regarding IFI/IFD after Car-T therapy. It is improved over the prior version.
Comments:
The manuscript has corrected many of the variable ways that they addressed genus and species names throughout, but there are still some errors.
- For example, when the abbreviation for species is used (“spp.”), it does not need to be italicized as it is not a Latin name.
- Another example, when “Cryptococci” is used, it is no longer the genus, it is the plural form of the organism, so it does not need to be in italics and it does not need to be capitalized.
- “Blastomyces dermatitidis” and “Pseudomonas aeruginosa” should be in italics.
When you state “In the study of Wittmann et al.,”, do you actually mean “In the study of Wittmann Dayagi et al.,”?
Author Response
Reviewer 1
Evangelidis and colleagues from Greece submit a revised manuscript regarding IFI/IFD after Car-T therapy. It is improved over the prior version.
Response: We would like to thank the reviewer for the time dedicated to reviewing our manuscript. Indeed, the comments and the corrections of the reviewer were substantial for the quality of our work. Please find below a point-to-point response to your comments. We remain at your disposal for anything that you might need.
Comments:
The manuscript has corrected many of the variable ways that they addressed genus and species names throughout, but there are still some errors.
- For example, when the abbreviation for species is used (“spp.”), it does not need to be italicized as it is not a Latin name.
Response: We are grateful for this comment. We have made changes throughout the manuscript.
- Another example, when “Cryptococci” is used, it is no longer the genus, it is the plural form of the organism, so it does not need to be in italics and it does not need to be capitalized.
Response: Thanks for this comment. We changed it to the correct form.
- “Blastomyces dermatitidis” and “Pseudomonas aeruginosa” should be in italics.
Response: The reviewer is right. We changed these phrases in their correct form.
When you state “In the study of Wittmann et al.,”, do you actually mean “In the study of Wittmann Dayagi et al.,”?
Response: We are sorry for this omission. We mean In the study of Wittmann Dayagi et al. The essential addition has been made in the revised version of the manuscript.
Reviewer 2 Report
Comments and Suggestions for Authors
fine for me
Author Response
We would like to thank the reviewer for the time dedicated to reviewing our manuscript. Indeed, the comments and the corrections of the reviewer were substantial for the quality of our work.